# LEARNING DYNAMIC STATE ABSTRACTIONS FOR MODEL-BASED REINFORCEMENT LEARNING

## ABSTRACT

A key challenge in model-based reinforcement learning (RL) is to synthesize computationally efficient and accurate environment models. We show that carefully designed models that learn predictive and compact state representations, also called state-space models, substantially reduce the computational costs for predicting outcomes of sequences of actions. Extensive experiments establish that state-space models accurately capture the dynamics of Atari games from the Arcade Learning Environment (ALE) from raw pixels. Furthermore, RL agents that use Monte-Carlo rollouts of these models as features for decision making outperform strong model-free baselines on the game MS_PACMAN, demonstrating the benefits of planning using learned dynamic state abstractions.

## 1    INTRODUCTION

Deep reinforcement learning has demonstrated remarkable progress in recent years, achieving high levels of performance across a wide array of challenging tasks, including Atari games (Mnih et al., 2015), locomotion (Schulman et al., 2015), and 3D navigation (Mnih et al., 2016). Many of these advances have relied on combining deep learning methods with model-free RL algorithms. A critical drawback of this approach is the vast amount of experience required to achieve good performance, as only weak prior knowledge is encoded in the agents' networks (e.g., spatial translation invariance via convolutions).

The promise of *model-based* reinforcement learning is to improve sample-efficiency by making use of explicit models of the environment. The idea is that given a model of the environment (which can possibly be learned in the absence of rewards or from observational data only), an agent can learn task-specific policies rapidly by leveraging this model e.g., by trajectory optimization (Betts, 1998), search (Browne et al., 2012; Silver et al., 2016a), dynamic programming (Bertsekas et al., 1995) or generating synthetic experiences (Sutton, 1991). However, model-based RL algorithms typically pose strong requirements on the environment models, namely that they make predictions about the future state of the environment *efficiently* and *accurately*.

Recent innovations combining model-free and model-based methods have helped to increase robustness to model imperfections. In the framework of Weber et al. (2017), the Imagination-Augmented Agent (I2A) queries its internal, *pre-trained model* via Monte-Carlo rollouts under a *rollout policy*. It then uses features computed from these rollouts to anticipate the outcomes of taking different actions, thereby informing its decision-making. RL is used to *learn to interpret* the model's predictions; this was shown to greatly diminish the susceptibility of planning to model imperfections. Nevertheless, training of efficient models and integrating them into the I2A architecture remains difficult. In fact, this challenge prevented the application of I2As to environments with complex transitions (Weber et al., 2017).

In this paper we address, the I2A framework, the main challenge posed by model-based RL: training accurate, computationally efficient models on more complex domains and using them with agents. First, we consider computationally efficient *state-space environment models* that make predictions at a higher level of abstraction, both spatially and temporally, than at the level of raw pixel observations. Such models substantially reduce the amount of computation required to perform rollouts, as future states can be represented much more compactly. Second, in order to increase model accuracy, we examine the benefits of *explicitly modeling uncertainty* in the transitions between these abstract states. Finally, we explore different strategies of *learning rollout policies* that define the interface

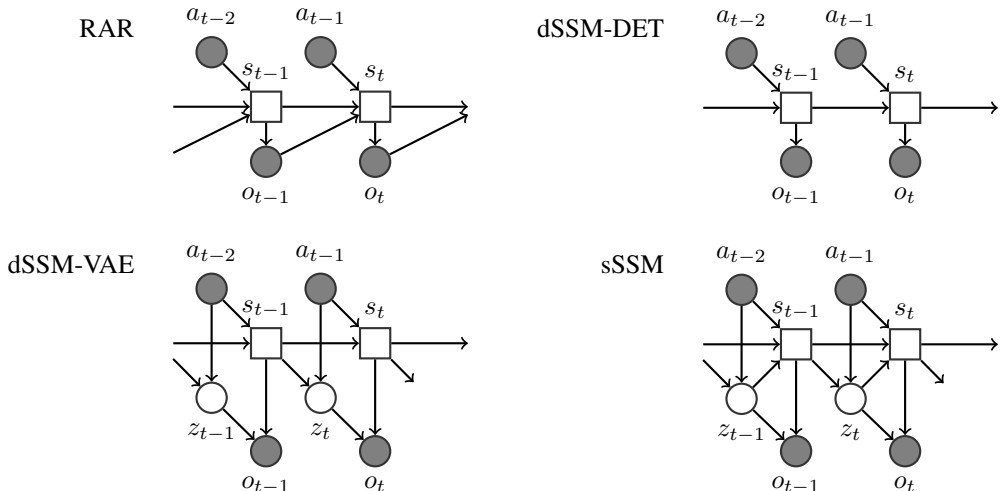

Figure 1: The graphical models representing the architectures of different environment models. Boxes are deterministic nodes, circles are random variables and filled circles represent variables observed during training.

between agent and environment model: We consider the possibility of *learning to query* the internal model, for guiding the Monte-Carlo rollouts of the model towards informative outcomes.

The main contributions of the paper are as follows: 1) we provide a comprehensive comparison of deterministic and stochastic, pixel-space and state-space models on a number of challenging environments from the Arcade Learning Environment (ALE, Bellemare et al., 2013); 2) we demonstrate state-of-the-art environment modeling accuracy (as measured by log-likelihoods) with stochastic, state-space models that efficiently produce diverse yet consistent rollouts; 3) using state-space models, we show model-based RL results on MS_PACMAN, and obtain significantly improved performance compared to strong model-free baselines, and 4) we show that learning to query the model further increases policy performance.

## 2 ENVIRONMENT MODELS

In the following, for any sequence of variables $x$, we use $x_{<t}$ (or $x_{\leq t}$) to denote all elements of the sequences up to $t$, excluding (respectively including) $x_t$. We write subsequences $(x_t, x_{t+1}, \ldots, x_s)$ as $x_{t:s}$. We denote the observations and rewards from the environment with $o_t$ and $r_t$. We also refer to the observations $o_t$ as *pixels* or *frames*, to give the intuition that they are high-dimensional and highly redundant in many domains of interest. To ease the notation, in the following we will also write $o_t$ for the observations and rewards $(o_t, r_t)$ unless explicitly stated otherwise.

Given action $a_t$, the environment transitions into a new state and returns a sample of the observation and reward of the next time step with probability $p^*(o_{t+1}|o_{\leq t}, a_{\leq t})$. A main challenge in model-based RL is to learn an accurate and efficient model $p$ of the environment $p^*$. Given a perfect model $p \approx p*$ and unlimited computational resources, an agent could e.g. perform in principle a brute-force search for the optimal open-loop policy $a^*_{t:T-1}$ in any state $o_{\leq t}$ by computing $\mathrm{argmax}_{a_{t:T-1}} \mathbb{E}_p[\sum_{s=t+1}^{T} r_s|o_{\leq t}, a_{<T})]$ (assuming undiscounted reward over a finite horizon up to $T$), where $\mathbb{E}_p$ is the expectation under the environment model $p$.

In practice, however, this optimization is costly and brittle. Given unavoidable imperfections of $p$ when modelling complex environments, it often leads to catastrophic planning outcomes (Talvitie, 2015). Therefore, we focus here on using models to just predict at time $t$ any future statistics $x_{t+1:t+\tau}$ over a horizon $\tau$ that are useful features for decision making. We call $x_{t+1:t+\tau}$ *imaginations* and $\tau$ the rollout horizon or depth. Concretely, we will assume that we are interested at every time step $t$ in generating samples $x_{t+1:t+\tau}$ by doing Monte-Carlo rollouts from the model $p$ given an arbitrary sequence of actions $a_{t:t+\tau-1}$ (which will later be sampled from the rollout policy).

## 2.1 MODEL TAXONOMY

In the following, we present different choices for the model $p$ and for which imagination statistic $x$ they compute (and subsequently pass to the agent). In particular, we focus on how these design choices trade-off computational efficiency with providing useful information to the agent. The structure of the models we consider are illustrated in Fig. 1.

### AUTO-REGRESSIVE MODELS

A straight-forward choice is the family of causal, temporally auto-regressive models over the observations $o_{t+1:t+\tau}$, which we write in the following way:

$$p(o_{t+1:t+\tau}|o_{\leq t}, a_{<t+\tau}) = \prod_{r=t+1}^{t+\tau} p(o_r|f(o_{<r}, a_{<r})).$$

If $f$ is given by a first-in-first-out (FIFO) buffer of the last $K$ observations and actions $o_{r-K:r-1}, a_{r-K:r-1}$, the above definition is a regular auto-regressive model (of order $K$), which we denote by AR. Rolling out AR models is slow for two reasons: 1) we have to sequentially sample, or "render", all pixels $o_{t+1:t+\tau}$ explicitly, which is particularly computationally demanding for high-dimensional observations, and 2) vanilla AR models without any additional structure do not reuse any computations from evaluating $p(o_r|f(o_{<r}, a_{<r}))$ for evaluating $p(o_{r+1}|f(o_{\leq r}, a_{\leq r}))$. To speed-up AR models, we address the latter concern by considering the following model variant: we allow $f$ to be a recurrent mapping that recursively updates sufficient statistics $h_r = f(h_{r-1}, a_{r-1}, o_{r-1})$, therefore reusing the previously computed statistics $h_{r-1}$. We call these models recurrent auto-regressive models (RAR); if $f$ is parameterized as a neural network, RARs are equivalent to recurrent neural networks (RNNs). Although faster, we still expect Monte-Carlo rollouts of RARs to be slow, as they still need to explicitly render observations $o_{t+1:t+\tau}$. For both auto-regressive models, the natural choice for imaginations $x_{t+1:t+\tau} = o_{t+1:t+\tau}$ are predicted frames.

### STATE-SPACE MODELS: ABSTRACTION IN SPACE

As discussed above, rolling out ARs is computationally demanding as it requires sampling, or "rendering" all observations $o_{t+1:t+\tau}$. Causal state-space models (SSM) circumvent this by positing that there is a compact *state* representation $s_t$ that captures essential aspects of the environment on an abstract level: it is assumed that $s_{t+1}$ can be "rolled out", i.e. predicted, from the previous state $s_t$ and action $a_t$ alone, without the help of previous pixels $o_{\leq t}$ or any action other than $a_t$: $p(s_{t+1}|s_{\leq t}, a_{<t+\tau}, o_{\leq t+\tau}) = p(s_{t+1}|s_t, a_t)$. Hence SSMs allow for the following factorization of the predictive distribution:

$$p(o_{t+1:t+\tau}|o_{\leq t}, a_{<t+\tau}) = \int \prod_{r=t+1}^{t+\tau} \Big( p(s_r|s_{r-1}, a_{r-1})p(o_r|s_r) \Big) p_{\text{init}}(s_t|o_{\leq t}, a_{<t}) ds_{t:t+\tau}.$$

For this model class, we choose the imaginations $x_{t+1:t+\tau} \sim p(s_{t+1:t+\tau}|o_{\leq t}, a_{<t+\tau})$ to be sampled from the model distribution over the state representation. This choice implies that imaginations are, by construction, sufficient to generate all possible predictions, do not require sampling pixel observations and that they live in a conveniently lower-dimensional space.

**Transition model** We consider two flavors of SSMs: deterministic SSMs (dSSMs) and stochastic SSMs (sSSMs). For dSSMs, the latent transition $s_{t+1} = g(s_t, a_t)$ is a deterministic function of the past, whereas for sSSMs we transition distributions $p(s_{t+1}|s_t, a_t)$ that explicitly model uncertainty over the state $s_{t+1}$. sSSMs are a strictly larger model class than dSSMs, and we illustrate their difference in capacity for modelling stochastic time-series in the Appendix. We parameterize sSSMs by introducing for every $t$ a latent variable $z_t$ whose distribution depends on $s_{t-1}$ and $a_{t-1}$, and by making the state a deterministic function of the past state, action, and latent variable:
$$z_{t+1} \sim p(z_{t+1}|s_t, a_t), \qquad s_{t+1} = g(s_t, a_t, z_{t+1}).$$

**Observation model** The observation model, or *decoder*, computes the conditional distribution $p(o_t|\cdot)$. It either takes as input the state $s_t$ (deterministic decoder), or the state $s_t$ and latent $z_t$ (stochastic decoder). For sSSMs, we always use the stochastic decoder. For dSSMs, we can use either the deterministic decoder (dSSM-DET), or the stochastic decoder (dSSM-VAE). The latter can capture joint uncertainty over pixels in a given observation $o_t$, but not across time steps. Further details can be found in section A.1.2 in the Appendix.

## 2.2 JUMPY MODELS: ABSTRACTION IN TIME

To further reduce the computational time required for sampling a rollout of horizon $\tau$, we also consider modelling environment transitions at a coarser time scale. To this end, we sub-sample observations by a factor of $c$, i.e. for $\tau' = \lfloor \tau/c \rfloor$, we replace sequences $(o_t, o_{t+1}, \ldots, o_{t+\tau})$, by the subsampled sequence $(o_t, o_{t+c}, o_{t+2c}, \ldots, o_{t+\tau'c})$. We "chunk" the actions by concatenating them into a vector $a_t \leftarrow (a_t^\top, \ldots, a_{t+c-1}^\top)^\top$, and sum the rewards $r_t \leftarrow \sum_{s=0}^{c-1} r_{t+s}$. We refer to models trained on data pre-processed in this way as *jumpy* models. Jumpy training is a convenient way to inject temporal abstraction over at a time scale $c$ into environment models. This approach allows us to further reduce the computational load for Monte-Carlo rollouts roughly by a factor of $c$.

## 2.3 MODEL ARCHITECTURES, INFERENCE AND TRAINING

Here, we describe the parametric architectures for the models described above. We discuss the architecture of the sSSM in detail, and then briefly explain the modifications of this model used to implement RARs and dSSMs.

The states $s_t$, latent variables $z_t$ and observations $o_t$ are all shaped as convolutional feature maps and are generated by *transition modules* $z_t \sim p(z_t|s_{t-1}, a_{t-1})$, $s_t = g(s_{t-1}, z_t, a_{t-1})$, and the *decoder* $o_t \sim p(o_t|s_t, z_t)$ respectively. All latent variables are constrained to be normal with diagonal covariances. All modules consist of stacks of convolutional neural networks with ReLU nonlinearities. The transition modules use size-preserving convolutions, the decoder, size-expanding ones. To overcome the limitations of small receptive fields associated with convolutions, for modelling global effects of the environment dynamics, we use pool-and-inject layers: they perform max-pooling over their input feature maps, tile the results and concatenate them back to the inputs. Using these layers we can induce long-range spatial dependencies in the state $s_t$. All modules are illustrated in detail in the Appendix.

We train the AR, RAR and dSSM-DET models by maximum likelihood estimation (MLE), i.e. by maximizing $L(\theta) = \log p_\theta(o_{1:T}|a_{0:T-1}, \hat{o}_0)$ over model parameters $\theta$, where $T = 10$ and $\hat{o}_0$ denotes some initial context (in our experiments $\hat{o}_0 := o_{-2:0}$). We initialize the state $p_{\text{init}}(s_0|\hat{o}_0)$ with a convolutional network including an observation encoder $e$. This encoder $e$ uses convolutions that reduce the size of the feature maps from the size of the observation to the size of the state.

For the models containing latent variables, i.e. dSSM-VAE and sSSM, we cannot evaluate $L(\theta)$ in closed form in general. We maximize instead the evidence-based lower bound $\text{ELBO}_q(\theta) \leq L(\theta)$, where $q$ denotes an approximate posterior distribution, which reads as follows for the sSSM:

$$\text{ELBO}_q(\theta) = \sum_{t=1}^{T} \mathbb{E}_q[\log p(o_t|s_t) + \log p(z_t|s_{t-1}, a_{t-1}) - \log q(z_t|s_{t-1}, a_{t-1}, o_{t:T})],$$

where $\theta$ now denotes the union of the model parameters and the parameters of $q$. Here, we used that the structure of the sSSM to assume without loss of generality that $q$ is Markovian in $(z_t, s_t)$ (see Krishnan et al. (2015) for an in depth discussion). Furthermore, we restrict ourselves to the filtering distribution $q(z_t|s_{t-1}, a_{t-1}, o_t)$, which we model as normal distribution with diagonal covariance matrix. We did not observe improvements in experiments by using the full smoothing distribution $q(z_t|s_{t-1}, a_{t-1}, o_{t:T})$. We share parameters between the prior and the posterior by making the posterior a function of the state $s_t$ computed by the transition module $g$, as follows:

$$z_{t+1} \sim q(z_t|s_t, a_t, o_{t+1}), \qquad s_{t+1} = g(s_t, z_{t+1}, a_t).$$

The posterior uses the observation encoder $e$ on $o_{t+1}$; the resulting feature maps are then concatenated to $s_t$, and a number of additional convolutions compute the posterior mean and standard deviation of $z_{t+1}$. For all latent variable models, we use the reparametrization trick (Kingma & Welling, 2013; Rezende et al., 2014) and a single posterior sample to obtain unbiased gradient estimators of the ELBO.

We can restrict the above sSSM to a dSSM-VAE, by not feeding samples of $z_t$ into the transition model $g$. To ensure a fair model comparison (identical number of parameters and same amount of computation), we numerically implement this by feeding the mean $\mu_t$ of $p(z_t|s_{t-1}, a_{t-1})$ into the transition function $g$ instead. If we also do not feed $z_t$ (but the mean $\mu_t$) into the decoder for rendering $o_t$, we arrive at the dSSM-DET, which does not contain any samples of $z_t$. We implement the RAR based on the dSSM-DET by modifiying the transition model to $s_{t+1} = g(s_t, \mu_{t+1}, a_t, e(o_t))$, where $e(\cdot)$ denotes an encoder with the same architecture as the one of sSSM and dSSM-VAE.

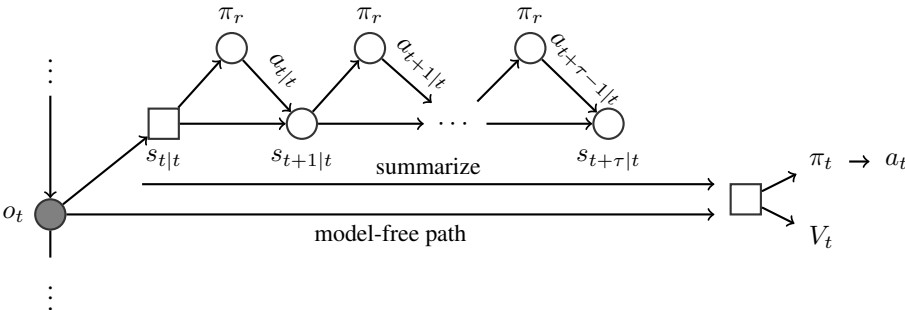

Figure 2: The architecture of the Imagination-Augmented Agent, which computes its policy $\pi_t$ and value function $V_t$, by combining information from a model-free path with information from Monte-Carlo rollouts of its environment model.

## 3  RL AGENTS WITH STATE-SPACE MODELS

We now use the environment models described above for model-based RL by integrating them into the Imagination-Augmented Agents (I2A). We briefly describe the agent, for details see Weber et al. (2017). The I2A is an RL agent with an actor-critic architecture, i.e. at each time step $t$, it explicitly computes its policy $\pi(a_t|o_{\leq t}, a_{<t})$ over the next action to take $a_t$ and an approximate value function $V(o_{\leq t}, a_{<t})$, and it is trained using standard policy gradient methods (Mnih et al., 2016). Its policy and value function are informed by the outputs of two separate pathways: 1) a model-free path, that tries to estimate the value and which action to take directly from the latest observation $o_t$ using a convolutional neural network (CNN); and 2) a model-based path, which we describe in the next paragraph.

The model-based path of an I2A is designed in the following way. The I2A is endowed with a *pre-trained, fixed* environment model $p$. At every time $t$, conditioned on past observations and actions $o_{\leq t}, a_{<t}$, it uses the model to simulate possible futures ("rollouts") represented by *imaginations* $x_{t+1:t+\tau}$ over a horizon $\tau$, under a *rollout policy* $\pi_r$. It then extracts informative features from the rollout imaginations $x$, and uses these, together with the results from the model-free path, to compute $\pi$ and $V$. It has been shown that I2As are robust to model imperfections: they learn to interpret imaginations produced from the internal models in order to inform decision making as part of standard return maximization. More precisely, the model-based path is computed by executing the following steps (also see Fig. 2):

- The I2A updates the state $s_t$ of its internal model by sampling from the initial model distribution $s_{t|t} \sim p_{\text{init}}(s_t|o_{\leq t})$. We denote this sample $s_{t|t}$ to clearly indicate the real environment information is contained in that sample up to time $t$.

- The I2A draws $K$ samples $x^{1:K}_{t+1:t+\tau|t}$ from the distribution $p_{\pi_r}(x_{t+1:t+\tau}|s_{t|t}, a_{\leq t})$. Here, $p_{\pi_r}$ denotes the model distribution with internal actions $a_{t:t+\tau|t}$ being sampled from the rollout policy $\pi_r$. For SSMs, we require the rollout policy to only depend on the state so that rollouts can be computed purely in abstract space.

- The imaginations $x^{1:K}_{t+1:t+\tau|t}$ are summarized by a "summarizer" module (e.g. an LSTM), then combined with the model-free output and finally used to compute $\pi(a_t|o_{\leq t}, a_{<t})$ and $V(o_{\leq t}, a_{<t})$.

Which imaginations $x$ the model predicts and passes to the agent is a design choice, which strongly depends on the model itself. As described above, for auto-regressive models (AR, RAR), we choose the imaginations to be rendered pixel predictions $o^k_{t+1:t+\tau|t}$. For SSM, we are free to use predicted pixels or predicted abstract states $s^k_{t+1:t+\tau|t}$ as imaginations, the latter being much cheaper to compute.

Apart from the choice of environment model, a key ingredient to I2As is the choice of internal actions applied to the model. How to best design a rollout policy $\pi_r$ that extracts useful information from a given environment model remains an open question, which also depends on the choice of model itself. In the following and we investigate in the following different possibilities.

## 3.1 DISTILLATION

In Weber et al. (2017), the authors propose to train the rollout policy $\pi_r$ to imitate the agent's model-based behavioral policy $\pi$. We call the resulting agent the *distillation agent*. Concretely, we minimize the Kullback-Leibler divergence between $\pi(\cdot|o_{\leq t}, s_{\leq t})$ and $\pi_r(\cdot|s_{t|t})$:

$$L_D[\pi_r] \quad = \quad \lambda_D \operatorname{KL}(\pi\|\pi_r) = -\lambda_D \mathbb{E}_\pi[\log \pi_r(a_t|s_{t|t})] + \text{const},$$

where $\mathbb{E}_\pi$ is the expectation over states and actions when following policy $\pi$. $\lambda_D$ is a hyperparameter that trades off reward maximization with the distillation loss.

## 3.2 LEARNING TO QUERY BY BACKPROPAGATION

An obvious alternative to distillation is to learn the parameters of $\pi_r$ jointly with the other parameters of the agents by policy gradient methods. As the rollout actions sampled from $\pi_r$ are discrete random variables, this optimization would require "internal" RL – i.e. redefining the action space to include the internal actions and learning a joint policy over external and internal actions. However, we expect the credit assignment of the rewards to the internal actions to be a difficult problem, resulting in slow learning. Therefore, we take a heurisitic approach similar to Henaff et al. (2017) (and related to Bengio et al., 2013): Instead of feeding the sampled one-hot environment action to the model, we can instead directly feed the probability vector $\pi_r(a_{t'}|s_{t'|t})$ into the environment model during rollouts. This can be considered as a *relaxation* of the discrete internal RL optimization problem. Concretely, we back-propagate the RL policy gradients through the entire rollout into $\pi_r$. This is possible since the environment model is fully differentiable thanks to the reparametrization trick, and the simulation policy is differentiable thanks to the relaxation of discrete actions. Parameters of the environment model $p$ are not optimized but kept constant, however. As the model was only trained on one-hot representation $a_t \in \{0, 1\}^N$, and not on continuous actions probabilities, it is not guaranteed a-priori that the model generalizes appropriately. We explore promoting rollout probabilities $\pi_r(\cdot|s_{t'|t})$ to be close to one-hot action vectors, and therefore are numerically closer to the training data of the model, by introducing an entropy penalty.

## 3.3 MODULATION AGENT

When learning the rollout policy (either by distillation or back-propagation), we learn to choose internal actions such that the simulated rollouts provide useful information to the agent. In these approaches, we do not change the environment model itself, which, by construction, aims to capture the true frequencies of possible outcomes. We can, however, go even one step further based on the following consideration: It might be beneficial for the agent to preferentially "imagine" extreme outcomes, e.g. rare (or even impossible) but highly rewarding or catastrophic transitions for a sequence of actions; hence to change the environment model itself in an informative way. For instance, in the game of MS_PACMAN, an agent might profit form imagining the ghosts moving in a particularly adversarial way, in order to choose actions safely. We can combine this consideration with the learning-to-query approach above, by learning an informative joint "imagination" distribution over actions and outcomes. We implement this in the following way. First, we train an *unconditional* sSSM on environment transitions, i.e. a model that does not depend on the executed actions $a_{<t}$ (this can simply be done by not providing the actions as inputs to the components of our state-space models). As a result, the sSSM has to jointly capture the uncertainty over the environment and the policy $\pi_{\text{data}}$ (the policy under which the training data was collected) in the latent variables $z$. This latent space is hence a compact, distributed representation over possible futures, i.e. trajectories, under $\pi_{\text{data}}$. We then let the I2A search over $z$ for informative trajectories, by replacing the learned prior module $p(z_t|s_{t-1})$ with a different distribution $p_{\text{imag}}(z_t|s_{t-1})$. The model is fully differentiable and we simply backpropagate the policy gradients through the entire model; the remaining weights of the model are left unchanged, except for those of $p_{\text{imag}}$. In our experiments, we simply replace the neural network parameterizing $p(z_t|s_{t-1})$ with a new one of the same size for $p_{\text{imag}}$, but with freshly initialized weights.

## 4 RESULTS

Here, we apply the above models and agents to domains from the Arcade Learning Environment (ALE, Bellemare et al., 2013). In spite of significant progress (Hessel et al., 2017), some games are still considered very challenging environments for RL agents, e.g. MS_PACMAN, especially when

| Model | BOWLING | CENTIPEDE | MS_PACMAN | SURROUND | rel. speed |
|---|---|---|---|---|---|
| AR | – | – | $1.9 \pm$ —— | – | $1.0\times$ |
| RAR | $-0.9 \pm 3.4$ | $\mathbf{5.6} \pm 0.3$ | $\mathbf{4.3} \pm 0.5$ | $-4.7 \pm 12.2$ | $2.0\times$ |
| dSSM-DET | $0.4 \pm 0.0$ | $3.5 \pm 0.2$ | $0.4 \pm 0.3$ | $-0.4 \pm 0.1$ | $5.2\times$ |
| dSSM-VAE | $0.5 \pm 0.0$ | $5.0 \pm 1.3$ | $2.4 \pm 3.0$ | $0.7 \pm 0.0$ | $5.2\times$ |
| sSSM | $\mathbf{0.6} \pm 0.0$ | $\mathbf{5.6} \pm 1.0$ | $\mathbf{4.3} \pm 0.3$ | $\mathbf{0.9} \pm 0.2$ | $5.2\times$ |
| sSSM (jumpy) | – | – | $3.0 \pm 2.0$ | – | $\mathbf{13.6}\times$ |

Table 1: Improvement of test likelihoods of environment models over a baseline model (standard variational autoencoder, VAE), on 4 different ALE domains. Stochastic models with state uncertainty (RAR, sSSM) outperform models without uncertainty representation. Furthermore, state-space models (dSSM, sSSM) show a substantial speed-up over auto-regressive models. Results are given as mean $\pm$ standard deviation, in units of $10^{-3} \cdot \mathrm{nats} \cdot \mathrm{pixel}^{-1}$.

not using any privileged information. All results are based on slightly cropped but full resolution ALE observations, i.e. $o_t \in [0,1]^{200 \times 160 \times 3}$.

## 4.1 COMPARISON OF ENVIRONMENT MODELS

We applied auto-regressive and state-space models to four games of the ALE, namely BOWLING, CENTIPEDE, MS_PACMAN and SURROUND. These environment where chose to cover a broad range of environment dynamics. The data was obtained by a running a pre-trained baseline policy $p_{\mathrm{data}}$ and collecting sequences of actions, observations and rewards $a_{1:T}, o_{1:T}, r_{1:T}$ of length $T = 10$. Results are computed on held-out test data. We optimized model hyper-parameters (learning rate, weight decay, mini-batch size) on one game (MS_PACMAN) for each model separately and report mean likelihoods over five runs with the best hyper-parameter settings. In Tab. 1, we report likelihood improvements over a baseline model, being a Variational Autoencoder (VAE) that models frames as independent (conditioned on three initial frames).

In general, we found that, although operating on an abstract level, SSMs are competitive with, or even outperform, auto-regressive models. The sSSM, which take uncertainty into account, achieves consistently higher likelihoods in all games compared to models with deterministic state transitions, namely dSSM-DET and dSSM-VAE, in spite of having the same number of parameters and operations. An example illustrating the differences in modelling capacity is shown in Fig. 17 (in the Appendix) on MS_PACMAN: the prediction of dSSM-DET exhibits "sprite splitting" (and eventually, "sprite melting") at corridors, whereas multiple samples from the sSSM show that the model has a reasonable and consistent representation of uncertainty in this situation.

We also report the relative computation time of rolling out, i.e. sampling from, the models. We observe that SSMs, which avoid computing pixel renderings at each rollout step, exhibit a speedup of $> 5$ over the standard AR model[1]. We want to point out that our AR implementation is already quite efficient compared to a naive one, as it reuses costly vision pre-processing for rollouts where possible. Furthermore, we show that a jumpy sSSM, which learns a temporally and spatially abstracted state representation, is faster than the AR model by more than an order of magnitude, while exhibiting comparable performance as shown in Tab. 1. This shows that using an appropriate model architecture, we can learn highly predictive and compact dynamic state abstractions. Qualitatively, we observe that the best models capture the dynamics of ALE games well, even faithfully predicting global, yet subtle effects such as pixel representation of games scores over tens of steps (see Fig. 17 in the Appendix).

## 4.2 RL AGENTS WITH STATE-SPACE MODELS ON MS_PACMAN

Here, we apply the I2A to a slightly simplified variant of the MS_PACMAN domain with five instead of eighteen actions. As environment models we use jumpy SSMs, since they exhibit a very favourable speed-accuracy trade-off as shown in the previous section; in fact I2As with AR models proved too expensive to run. In the following we compare the performance of I2As with different variants of SSMs, as well as various baselines. All agents we trained with an action repeat of 4

---

[1]For lack of time, we could not collect performance of the AR model on all games; this will be fixed in a later revision.

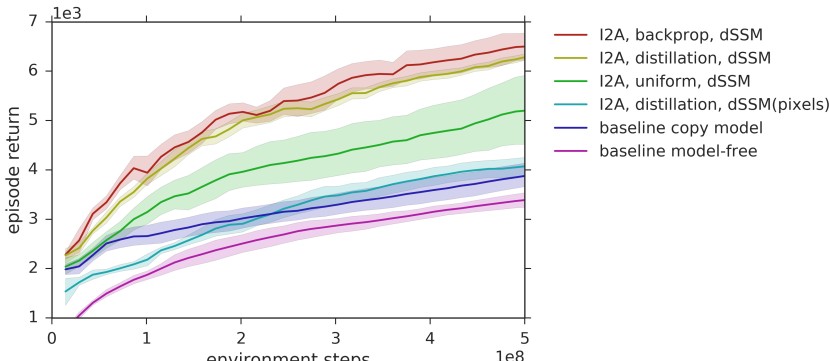

Figure 3: Learning curves of different agents on the MS_PACMAN environment. Model-based Imagination-Augmented Agents (I2As) outperform the model-free baseline by a large margin. Furthermore, learning the rollout policy $\pi_r$, either by back-propagation or distillation provides the best results.

(Mnih et al., 2015). We report results in terms of averaged episode returns as a function of experience (in number of environment steps), averaged over the best hyper-parameter settings. All I2As do $K = 5$ (equal to the number of actions) rollouts per time step. Rollout depth $\tau$ was treated as a hyper-parameter and varied over $\tau \in \{2, 3, 4\}$; this corresponds to 24, 36 and 48 environment steps (due to action repeats and jumpy training), allowing I2As to plan over a substantial horizon. Learning curves for all agents with deterministic dSSMs are shown in Fig.3. Results and detailed discussion for agents with sSSMs can be found in section A.3 the Appendix.

We first establish that all I2A agents, irrespective of the models they use, perform better than the model-free baseline agent; the latter is equivalent to an I2A without a model-based pathway. The improved performance of I2As is not simply due to having access to a larger number of input features: an I2A agent with an untrained environment model performs substantially worse (data not shown). A final baseline consists in using an I2A agent for which all imaginations $s_{t+1:t+\tau|t}$ are set to the initial state representation $s_{t|t}$. The agent has the exact same architecture, number of weights (forward model excluded), and operations as the I2A agent (denoted as "baseline copy model" in the figure legend). This agent performs substantially worse than the I2A agent, showing that environment rollouts lead to better decisions. It performs better however than the random model agent, which suggests that simply providing the initial state representation to the agent is already beneficial, emphasizing the usefulness of abstract dynamic state representations.

A surprising result is that I2As with the deterministic state-space models dSSM outperform their stochastic counterparts with sSSMs by a large margin. Although sSSMs capture the environment dynamics better than dSSM, learning from their outputs seems to be more challenging for the agents. We hypothesize that this could be due to the fact that we only produce only a small number of samples (5 in our simulations), resulting in highly variable features that are passed to the I2As.

For the agents with deterministic models, we find that a uniform random rollout policy is a strong baseline. It is outperformed by the distillation strategy, itself narrowly outperformed by the learning-to-query strategy. This demonstrates that "imagining" behaviors different from the agents' policy can be beneficial for planning. Furthermore, we found that in general deeper rollouts with $\tau = 4$ proved to outperfrom more shallow rollouts $\tau = 2, 3$ for all I2As with deterministic SSMs.

A final experiment consists of running the I2A agent with distillation, but instead of providing the abstract state features $s_{t+1:t+\tau|t}$ to the agent, we provide rendered pixel observations $o_{t+1:t+\tau|t}$ instead (as was done in Weber et al., 2017), and strengthen the summarizer (by adding convolutions). This model has to decode and re-encode observations at every imagination step, which makes it our slowest agent. We find that reasoning at pixel level eventually outperforms the copy and model-free baselines. It is however significantly outperformed by all variants of I2A which work at the abstract level, showing that the dynamics state abstractions, learned in an unsupervised way by a state-space model, are highly informative features about future outcomes, while being cheap to compute at the same time.

## 5 RELATED WORK

**Generative sequence models**   We build directly on a plethora of recent work exploring the continuum of models ranging from standard recurrent neural networks (RNNs) to fully stochastic models with uncertainty (Chung et al., 2015; Archer et al., 2015; Fraccaro et al., 2016; Krishnan et al., 2015; Gu et al., 2015). Chung et al. (2015) explore a model class equivalent to what we called RARs here. Archer et al. (2015); Fraccaro et al. (2016) train stochastic state-space models, without however investigating their computational efficiency and their applicability to model-based RL. Most of the above work focuses on modelling music, speech or other low-dimensional data, whereas here we present stochastic sequence models on high-dimensional pixel-based observations; noteworthy exception are found in Watter et al. (2015); Wahlström et al. (2015). There, the authors chose a two-stage approach by first learning a latent representation and then learning a transition model in this representation. Multiple studies investigate the graphical-model structure of the prior and posterior graphs and stress the possible importance of smoothing over filtering inference distributions (e.g. Krishnan et al., 2015); in our investigations we did not find a difference between these distributions. Furthermore, to the best our knowledge, this is the first study applying stochastic state-space models as action-conditional environment models. Most previous work on learning simulators for ALE games apply deterministic models, and do not consider learning state-space models for efficient Monte-Carlo rollouts (Oh et al., 2015). Chiappa et al. (2017) successfully train deterministic state-space models for ALE modelling (largely equivalent to the considered dSSMs here); they however do not explore the computational complexity advantage of SSMs, and do not study RL applications of their models.

**Model-based reinforcement learning**   Most model-based RL with neural network models has previously focused on training the models on a given, compact state-representations. Directly learning models from pixels for RL is still an under-explored topic due to high demands on model accuracy and computational budget, but see Finn & Levine (2017); Watter et al. (2015); Wahlström et al. (2015). Finn & Levine (2017) train an action-conditional video-prediction network and use it for model-predictive control (MPC) of a robot arm. The applied model requires explicit pixel rendering for long-term predictions and does not operate in abstract space. Agrawal et al. (2016) propose to learn a forward and inverse dynamics model from pixels with applications to robotics. Our work is related to multiple approaches in RL which aim to *implicitly* learn a model on the environment using model-free methods. Tamar et al. (2016) propose an architecture that is designed to learn the value-iteration algorithm which requires knowledge about environment transitions. The Predictron is another implicit planning network, trained in a supervised way directly from raw pixels, mimicking Bellman updates / iterations (Silver et al., 2016b). A generalization of the Predictron to the controlled setting was introduced by Oh et al. (2017). Another approach, presented by Jaderberg et al. (2016), is to add auxiliary prediction losses to the RL training criterion in order to encourage implicit learning of environment dynamics. van Seijen et al. (2017) obtain state of the art performance on MS_PACMAN with a model free architecture, but they however rely on privileged information (object identity and positions, and decomposition of the reward function).

## 6 DISCUSSION

We have shown that state-space models directly learned from raw pixel observations are good candidates for model-based RL: 1) they are powerful enough to capture complex environment dynamics, exhibiting similar accuracy to frame-auto-regressive models; 2) they allow for computationally efficient Monte-Carlo rollouts; 3) their learned dynamic state-representations are excellent features for evaluating and anticipating future outcomes compared to raw pixels. This enabled Imagination-Augmented Agents to outperform strong model-free baselines. On a conceptual level, we present (to the best of our knowledge) the first results on what we termed *learning-to-query*: We show a learning a rollout policy by backpropagating policy gradients leads to consistent (if modest) improvements.

Here, we adopted the I2A assumption of having access to a pre-trained envronment model. In future work, we plan to drop this assumption and jointly learn the model and the agent. Also, further speeding up environment models is a major direction of research; we think that learning models with the capacity of learning adaptive temporal abstractions is a particularly promising direction for achieving agents that plan to react flexibly to their environment.

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

## A APPENDIX

### A.1 DETAILS ON ENVIRONMENT MODELS

#### A.1.1 ARCHITECTURES

We show the structures the inference distributions of the models with latent variables in Fig. 4 and Fig. 5. .

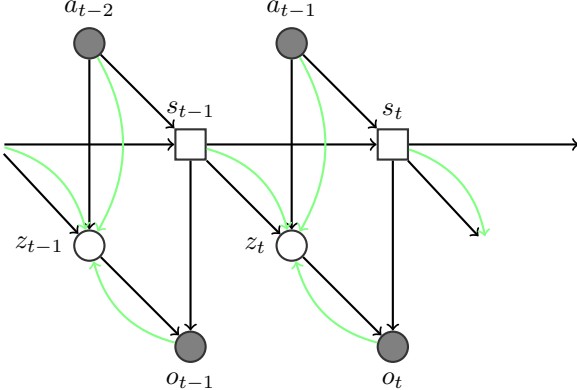

Figure 4: The architecture of the inference model $q$ for the dSSM-VAE.

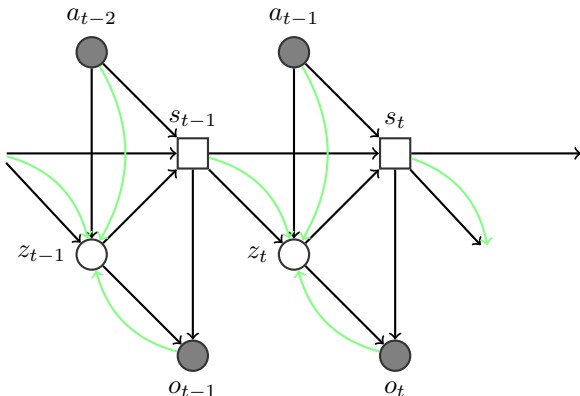

Figure 5: The architecture of the inference model $q$ for the sSSM.

#### A.1.2 DETAIL IN THE OBSERVATION MODEL

For all models (auto-regressive and state-space), we interpret the three color channels of each pixel in the observation $o_t \in [0, 1]^{H \times W \times 3}$ (with frame height $H$ and width $W$) as pseudo-probabilities; we score these using their KL divergence with model predictions. We model the reward with a separate distribution: we first compute a binary representation of the reward $\sum_{n=0}^{N-1} b_{t,n} 2^n = \lfloor r_t \rfloor$ and model the coefficients $b_{t,n}$ as independent Bernoulli variables (conditioned on $s_t, z_t$). We also add two extra binary variables: one for the sign of the reward, and the indicator function of the reward being equal to $0$.

#### A.1.3 DETAILS OF NEURAL NETWORK IMPLEMENTATIONS

Here we show the concrete neural network layouts used to implement the sSSM. We first introduce three higher level build blocks:

- a three layer deep convolutional stack $\mathrm{conv\_stack} : (k_i, c_i)_{i=1,2,3}$, with kernel sizes $k_1, k_2, k_3$ and channels sizes $c_1, c_2, c_3$, shown in Fig. 6;
- a three layer deep residual convolutional stack $\mathrm{res\_conv}$ with fixed sizes, shown in Fig. 7;
- the Pool & Inject layer, shown in Fig. 8.

Based on these building blocks, we define all modules in Fig. 9 to Fig. 14.

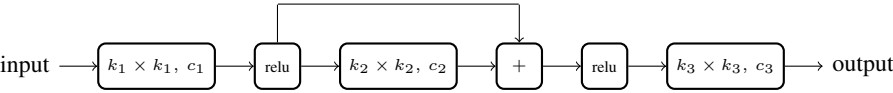

Figure 6: Definition of the basic convolutional stack $\mathrm{conv\_stack} : (k_i, c_i)_{i=1,2,3}$ with kernel size parameters $k_{1,2,3}$ and channel parameters $c_{1,2,3}$. Here, a box with the label $k_i \times k_i, c_i$ denotes a convolution with a square kernel of size $k_i$ with $c_i$ output channels; strides are always $1 \times 1$.

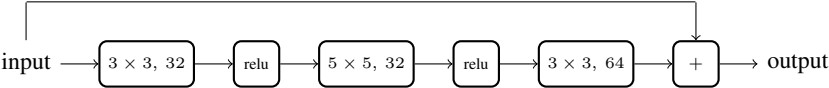

Figure 7: Definition of the residual convolutional stack $\mathrm{res\_conv}$.

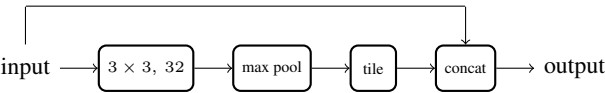

Figure 8: Definition of the Pool & Inject layer.

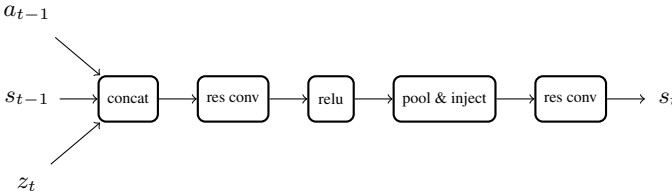

Figure 9: Transition module for computing the state transition function $s_t = g(s_{t-1}, z_t, a_{t-1})$.

### A.1.4 COLLECTION OF TRAINING DATA

We train a standard DQN agents on the four games BOWLING, CENTIPEDE, MS PACMAN and SURROUND from the ALE as detailed by Mnih et al. (2015) using the original action space of 18 actions. After training, we collect a training set of $10^8$ and a test set of $10^7$ environment transitions for each game by executing the learned policies. Actions are represented by one-hot vectors and are tiled to yield convolutional feature maps of appropriate size. Pixel observations $o_t$ were cropped to $200 \times 160$ pixels and normalized by 255 to lie in the unit cube $[0, 1]^3$. Because the DQN agent were trained with an action-repeat of four, we only model every fourth frame.

### A.1.5 TRAINING DETAILS

All models were optimized using Adam (Kingma & Ba, 2014) with a mini-batch size of 16.

### A.1.6 COMPARISON OF DETERMINISTIC AND STOCHASTIC STATE-SPACE MODELS

We illustrate the difference in modelling capacity between deterministic (dSSM) and stochastic (sSSM) state-space models, by training both on a toy data set. It consists of small $80 \times 80$-pixel image sequences of a bouncing ball with a drift and a small random diffusion term. As shown in

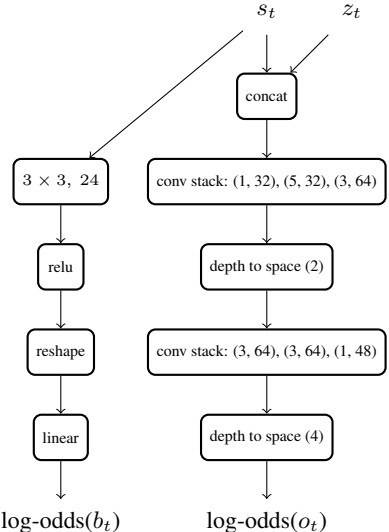

Figure 10: Decoder module for computing the log-odds statistics of the Bernoulli distributions over the pixels $o_t$ and the binary coefficients of the reward $\lfloor r_t \rfloor = \sum_{n=0}^{N-1} b_{t,n} 2^n$.

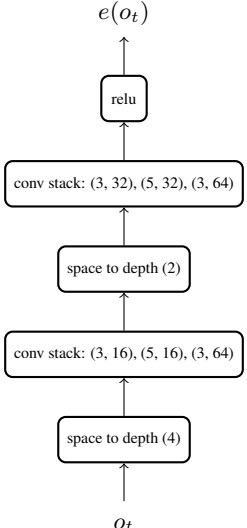

Figure 11: Encoder module computing an embedding $e(o_t)$ of an observation $o_t$ (not including the reward).

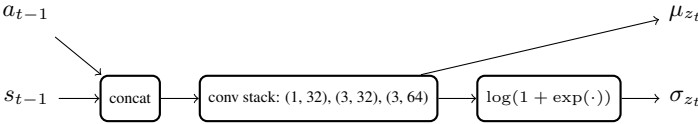

Figure 12: Prior module for computing mean $\mu_{z_t}$ and diagonal variance $\sigma_{z_t}$ of the normal distribution $p(z_t | s_{t-1}, a_{t-1})$.

Fig. 16, after training, pixels rendered from the rollouts of a sSSM depict a plausible realization of a trajectory of the ball, whereas the dSSM produces blurry samples, as conditioned on any number of previously observed frames, the state of the ball is not entirely predictable due to diffusion. A

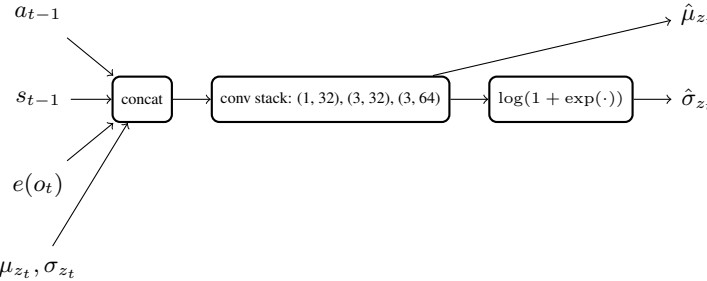

Figure 13: Posterior module for computing mean $\hat{\mu}_{z_t}$ and diagonal variance $\hat{\sigma}_{z_t}$ of the normal distribution $q(z_t|s_{t-1}, a_{t-1}, o_t)$. The posterior gets as additional inputs the prior statistics $\mu_{z_t}, \sigma_{z_t}$.

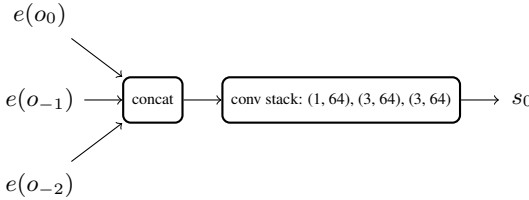

Figure 14: Initial state module for computing the first initial state $s_0$ as a function of the embedding $e(o_i)$ for $i = -2, -1, 0$ of three previous observations.

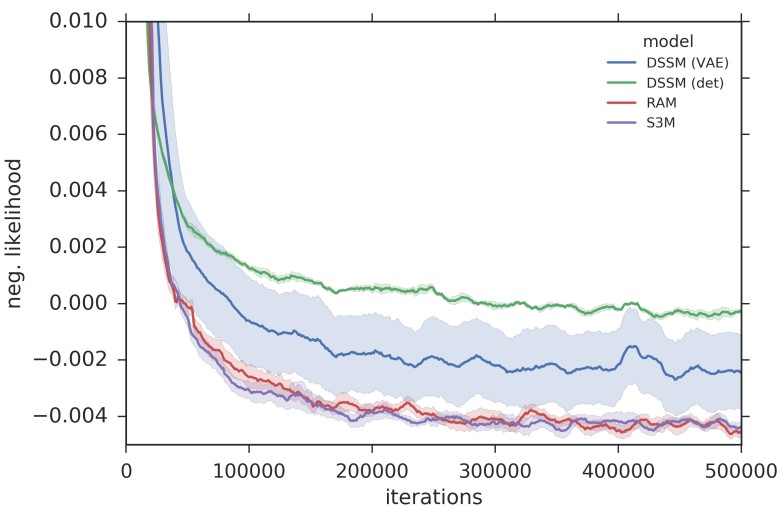

Figure 15: Learning curves of the environment models on MS_PACMAN.

dSSM (trained with an approximate maximum likelihood criterion, see above) will "hedge its bets" by producing a blurry prediction. A similar result can be observed in rollouts from models trained on ALE games, see Fig. 17.

## A.2 APPENDIX: DETAILS ON AGENTS

### A.2.1 MS PACMAN ENVIRONMENT VARIANT

For the RL experiments in the paper, we consider a slightly simplified version of the MS PACMAN environment with only five actions (UP, LEFT, DOWN, RIGHT, NOOP). Furthermore, all agents have an action-repeat of four, and only observe every fourth frame from the environment.

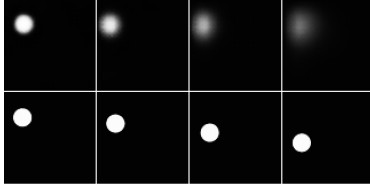

Figure 16: Rollouts from a deterministic (dSSM, above) and a stochastic (sSSM, below) state-space model trained on a bouncing ball dataset with diffusion.

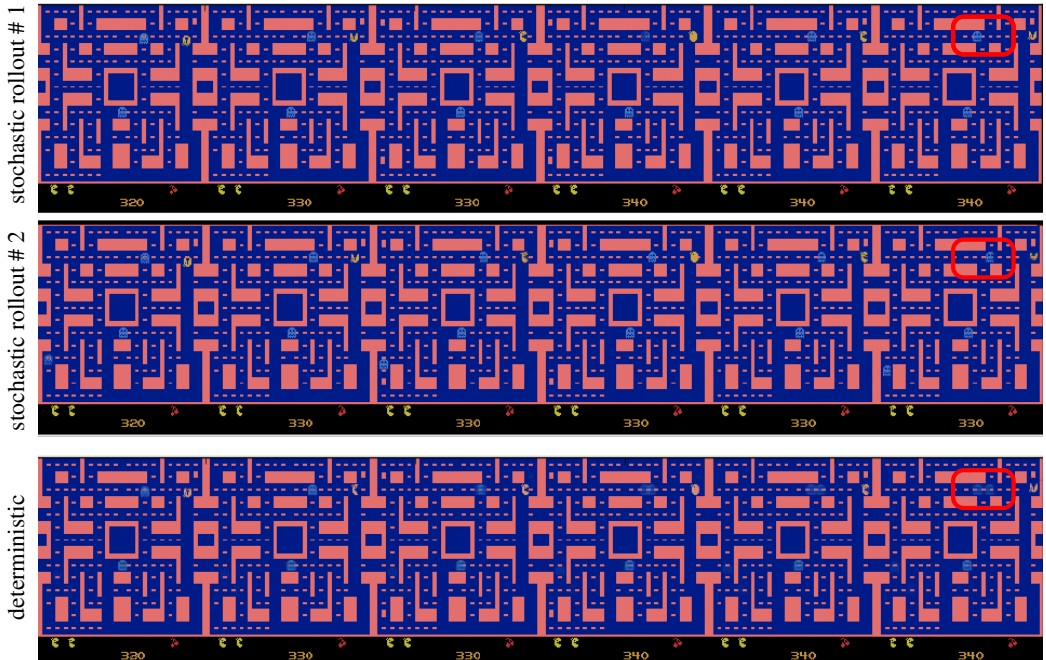

Figure 17: Two rollouts of length $\tau = 6$ from a stochastic (sSSM, top two rows) and one rollout from a deterministic (dSSM) state-space model for the MS PACMAN environment, given the same initial frames and the same sequence of five actions.

### A.2.2 ARCHITECTURE

We re-implemented closely the agent architecture presented by Weber et al. (2017). In the following we list the changes in the architecture necessitated by the different environments and environment models.

**Model-free baseline** The model-free baseline consisted of a four-layer CNN operating on $o_t$ with sizes (4, 2, 16), (8, 4, 32), (4, 2, 64) and (3, 1, 64), where $(k, s, c)$ donates a CNN layer with square kernel size $k$, stride $s$ and output channels $s$; each CNN layer is followed by a relu nonlinearity. The output of the CNN is flatten and passed trough a fully-connected (FC) layer with 512 hidden units; the final output is a value function approximation and the logits of the policy at time $t$.

**Imagination-Augmented Agent (I2A)** The model-free path consists of a CNN with the same size as the one of the model-free agent (including the FC layer with 512 units). The model-based path is designed as follows: The rollout outputs for each imagined time step $s$ are encoded with a two layer CNN with sizes (4, 1, 32) and (4, 1, 16), then flattened and passed to a fully-connected (FC) layer with 128 outputs. These rollout statistics are then summarized (in reversed order) with an LSTM with 256 hidden units and concatenated with the outputs of the model-free path.

**Rollout policies** Trainable rollout policies that operate on the state $s_t$ are given by a two layer CNN with sizes (4, 1, 32) and (4, 1, 32), followed by an FC layer with 128 units. Pixel-based rollout policies have the same neural network sizes as the model-free baseline, except that the last two CNN layers have 32 feature maps each.

### A.3 RESULTS I2A WITH STOCHASTIC STATE-SPACE MODELS

Learning curves for I2As with sSSMs are shown in Fig. 18. Both, I2As with learing-to-query and distillation rollout policies outperform a uniform random rollout policy. The learning-to-query agent shows weak initial performance, but eventually outperforms the other agents. This shows that *learning-to-sample* informative outcomes is beneficial for agent performance.

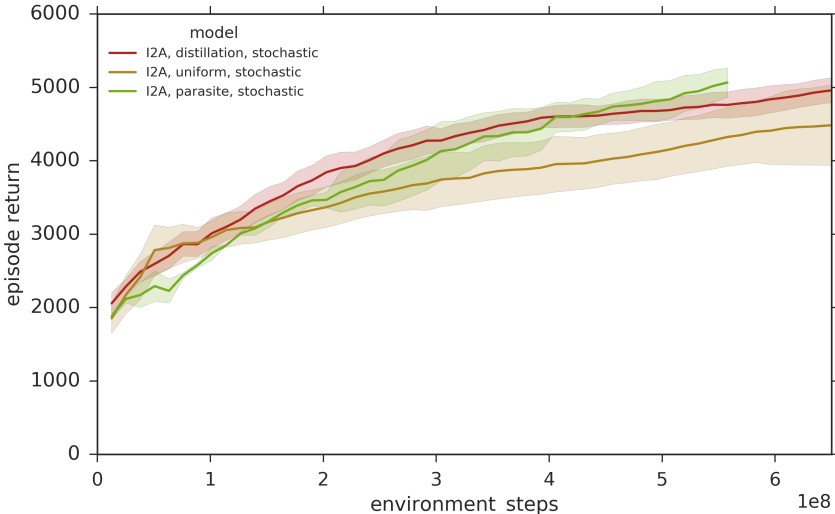

Figure 18: Model results.

