# OpenReview forum: "Learning Dynamic State Abstractions for Model-Based Reinforcement Learning"
_ICLR.cc/2018/Conference — Reject_

### Official Review · AnonReviewer2 · 2017-11-27
**Simple idea that seems to work well on Ms Pacman, but paper needs more quantitative results/qualitative inspections**

**Rating:** 6
**Confidence:** 4

**Review:**

Summary:

This paper studies how to learn (hidden)-state-space models of environment dynamics, and integrate them with Imagination-Augmented Agents (I2A). The paper considers single-agent problems and tests on Ms Pacman etc.

There are several variations of the hidden-state space [ds]SSM model: using det/stochastic latent variables + using det/stochastic decoders. In the stochastic case, learning is done using variational methods.

[ds]SSM is integrated with I2A, which generates rollouts of future states, based on the inferred hidden states from the d/sSSM-VAE model. The rollouts are then fed into the agent's policy / value function.

Main results seem to be:
1. Experiments on learning the forward model, show that latent forward models work better and faster than naive AR models on several Atari games, and better than fully model-free baselines.
2. I2A agents with latent codes work better than model-free models or I2A from pixels. Deterministic latent models seem to work better than stochastic ones.

Pro:
- Relatively straightforward idea: learn the forward model on hidden states, rather than raw states.
- Writing is clear, although a bit dense in places.

Con:
- Paper only shows training curves for MS Pacman. What about the other games from Table 1?
- The paper lacks any visualization of the latent codes. What do they represent? Can we e.g. learn a raw-state predictor from the latent codes?
- Are the latent codes relevant in the stochastic model? See e.g. the discussion in "Variational Lossy Autoencoder" (Chen et al. 2016)
- Experiments are not complete (e.g. for AR, as noted in the paper).
- The games used are fairly reactive (i.e. do not require significant long-term planning), and so the sequential hidden-state-space model does not have to capture long-term dependencies. It would be nice to see how this technique fares on Montezuma's revenge, for instance.

Overall:
The paper proposes a simple idea that seems to work well on reactive 1-agent games. However, the paper could give more insights into *how* this works: e.g. a better qualitative inspection of the learned latent model, and how existing questions surrounding sequential stochastic model affect the proposed method. Also, not all baseline experiments are done, and the impact on training is only evaluated on 1 game.

Detailed:
-

---

> ### Author Response · Authors · 2017-12-14
> **Re: Simple idea...**
>
> We thank the reviewer for the comments. We would like to point out that another important contribution is that we consider learning the rollout policy by backpropagation of policy gradients via continuous relaxation an important point of the paper.
>
>
> "- Paper only shows training curves for MS Pacman. What about the other games from Table 1?"
>
> In spite of the 13x speed-up we achieved for the models, the main limitation of the imagination-augmented agent is that it is still expensive compared to the baseline. Therefore, we did not manage to do extensive experiments on the other games, and focused on MS_PACMAN, which is the most challenging one of the 4 games (based on DQN and A3C results). Once we finish developing our simplified experimentation pipeline with online-learning of the models, we will perform experiments on a wide range of domains.
>
> "- Experiments are not complete (e.g. for AR, as noted in the paper)."
>
> Point taken. We are currently re-running the experiments to complete the table.
>
>
> "- The paper lacks any visualization of the latent codes. What do they represent? Can we e.g. learn a raw-state predictor from the latent codes?"
>
> Although very interesting, due to space constraints (and considering that the paper is already quite dense), we did not include an analysis of the latent codes.
>
>
> “- Are the latent codes relevant in the stochastic model? See e.g. the discussion in "Variational Lossy Autoencoder" (Chen et al. 2016)”
>
> We found the latent variables to be relevant in the sense that they boost performance. As shown in Table 1, a fully deterministic model with the same number of parameters and operations but without latent variables (dSSM-DET), performs worse that the full state-space model (sSSM). Preliminary experiments also suggested that there is a  sweet spot for the ratio of deterministic hidden units vs latent variables (at fixed total size) for the state-space model.
>
>
> "- The games used are fairly reactive (i.e. do not require significant long-term
> planning), and so the sequential hidden-state-space model does not have to
> capture long-term dependencies. It would be nice to see how this technique fares
> on Montezuma's revenge, for instance"
>
> We agree that more experiments are necessary for fully assessing the benefits and limitations of the I2A, and we are currently working on this. However, our contribution is somewhat orthogonal to hard exploration problems like MONTEZUMAS_REVENGE, and we actually do not expect massive benefits in this particular domain (unless one uses model uncertainty for exploration see e.g. “Unifying count-based exploration and intrinsic motivation”. Bellemare NIPS 2016). Also, although MS_PACMAN is almost fully observed and exploration is not hard, is is a difficult domain in which standard model-free agents have underperformed. It is not fully reactive in a sense that planning of the order of tens of steps into the future seems beneficial.

---

### Official Review · AnonReviewer1 · 2017-12-04
**Unclear contribution**

**Rating:** 5
**Confidence:** 4

**Review:**

The paper proposes a method for inferring dynamical models from partial observations, that can later be used in model-based RL algorithms such as I2A. The essence of the method is to perform variational inference of the latent state, representing its distribution as Gaussian, and to use it in an ELBO to infer the dynamics of state and observation.

While this is an interesting approach, many of the architectural choices involved seem arbitrary and unjustified. This wouldn't be so bad if they were justified by empirical success rather than principled design, but I'm also a bit skeptical of the strength of the results.

A few examples of such architectural choices:
1. What's the significance of separating the stochastic state transition into a stochastic choice of z and a transition g deterministic in z?
2. How is the representational power affected by having only the observations depend on z? What's the intuition behind calling this model VAE, when sSSM is also trained with variational inference?
3. What is gained by using pool-and-inject layers? By the way, is this a novel component? If so please elaborate, if not please cite.

As for the strength of the results, in Table 1 the proposed methods don't seem to outperform "RAR" (i.e., RNN) in expected value. They do seem to have lower variance, and the authors would do well to underline the importance of this.
In Figure 3, it's curious that the model-free baseline remains unnamed, as it also does in the text and appendix. This makes it hard to evaluate whether the significant wins are indicative of the strength of the proposed method.

Finally, a notational point that the authors should really get right, is that conditioning on future actions naively changes the distribution of the current state or observation in ways they didn't intend. The authors intended for actions to be used as "interventions", i.e. a-causally, and should denote this conditioning by some sort of "do" operator.

---

> ### Author Response · Authors · 2017-12-14
> **Clarification of comments needed; part 1**
>
> "This wouldn't be so bad if they were justified by empirical success rather than principled design, but I'm also a bit skeptical of the strength of the results."
>
> We remain convinced that our results are strong. We would like to invite the reviewer to provide references that lead her / him to be sceptical of the strength of the results.
>
> Regarding our reinforcement learning results: To the best of our knowledge, we are the first to present model-based RL agents on ALE domains, where the model is learned from raw pixels without any privileged information. Our agents outperform a strong, standard A3C baseline (also see below).
>
> Regarding the unsupervised modelling results: As shown in Table 1, our proposed model achieves the best likelihoods on all games we tried compared to the baselines, while showing an approximately 3x speed-up over an RNN. In practice this means that an imagination-augmented agent can be trained in 4 days instead of say 2 weeks. Furthermore, using the jumpy sSSM we can train an I2A in **1 day** at comparable accuracy. Furthermore, to the best of our knowledge, we are the first to present results of stochastic sequence models on the Atari domain. Prior work on Atari was based on deterministic models (with their inherent limitations); prior work on stochastic models was limited to much lower-dimensional sequences.
>
>
> "While this is an interesting approach, many of the architectural choices involved seem arbitrary and unjustified."
>
> All architectural choices are informed by state-of-the-art generative sequence models cited in the paper. We actually tried to go for the simplest architectures wherever possible, most of which are standard, even in the wider context of deep learning, as detailed below.
>
>
> "1. What's the significance of separating the stochastic state transition into a stochastic choice of z and a transition g deterministic in z?"
>
> Instead of being idiosyncratic, this architectural choice is canonical: it directly follows from the standard decomposition of the joint probability $p(z_1,..., z_T)=\prod_t p(z_t\vert z_{<t})$. The sufficient statistics of the distribution over $z_t$ is a (deterministic) function of $z_{<0}=z_1,...,z_{t-1}$; let’s denote it by h_{t-1}. Assuming a large enough number of hidden units, an RNN is a general function approximator, and so we can use it to approximate $h_{t-1}$. This is exactly the architecture we use. For a similar line of argumentation, see e.g. Chen et al “Variational Lossy Autoencoder” (ICLR 2017, page 3 top) and references therein.
> Furthermore, this architecture is standard in the literature at least since [Chung et al. NIPS 2015] (e.g. see their Figure 1). Our own preliminary experiments also show empirically, that models with this architecture outperform models with purely stochastic hidden units.
>
>
> "2. How is the representational power affected by having only the observations depend on z?"
>
> We actually briefly discuss this difference in A.1.6 (page 13), which we explicitly reference to in the main article.

---

> > ### Author Response · Authors · 2017-12-14
> > **Clarification of comments needed; part 2**
> >
> > "What's the intuition behind calling this model VAE, when sSSM is also trained with variational inference?"
> >
> > This name was chosen, as the observation model of the dSSM-VAE is really a (conditional) convolutional, variational auto-encoder. The sSSM is indeed also trained by ELBO maximization, so in this sense it is also a VAE. However, it has additional, very important temporal structure, which we explicitly leverage for efficient inference and parameter optimization. We chose to call it a state-space model to emphasize this structure.
> >
> >
> > "3. What is gained by using pool-and-inject layers? By the way, is this a novel component? If so please elaborate, if not please cite."
> >
> > The reviewer is correct: this layer was already used in [Weber et al. NIPS2017], we will make this explicit. As stated in the manuscript, this layer makes it possible to capture spatial long-range dependencies in $s_t$. The state-transitions are modeled with convolutions. The size of the convolutional filters limits how "far" information can propagate spatially in a single state transition. Modelling global aspects of the environment state, eg the score / reward in MS_PACMAN, would require very large (and therefore costly) convolutional filters. The global pooling in the pool-and-inject layer fixes this. An alternative would have been eg to use large dilated convolutions.
> >
> >
> > "As for the strength of the results, in Table 1 the proposed methods don't seem to outperform "RAR" (i.e., RNN) in expected value..”
> >
> > See comment above.
> >
> >
> > "In Figure 3, it's curious that the model-free baseline remains unnamed, as it also does in the text and appendix. This makes it hard to evaluate whether the significant wins are indicative of the strength of the proposed method."
> >
> > Although it is mentioned in the manuscript, we agree that this could be somewhat confusing and it will be clarified in the next revision. The baseline is a re-implementation of the A3C agent from “Asynchronous Methods for Deep Reinforcement Learning” (Mnih et al ICLM 2016). This agent achieved state-of-the-art results on ALE, and is an extremely widely applied, general and strong baseline.
> >
> >
> > "Finally, a notational point that the authors should really get right, is that conditioning on future actions naively changes the distribution of the current state or observation in ways they didn't intend. The authors intended for actions to be used as "interventions", i.e. a-causally, and should denote this conditioning by some sort of "do" operator."
> >
> > The reviewer is incorrect about this point. As can be seen from Figure 1, the actions that we condition on, do not have any parents in the graphical models (because we deliberately did not include the policy for modelling the actions). Therefore, inference in the mutilated graph stemming from an intervention on the actions, is exactly equivalent to the conditional distribution stated on page 3.
> > Of course, instead of giving the conditional distribution on page 3, we could have used the do-notation. We decided against this, as it is rarely used (and could be therefore confusing) in the generative modelling and reinforcement literature, if it is not absolutely necessary.

---

### Official Review · AnonReviewer3 · 2017-12-05
**Could be impactful**

**Rating:** 8
**Confidence:** 4

**Review:**

The authors provide a deeper exploration of Imagination Agents, by looking more closely at a variety of state-space models.  They examine what happens as both the representation of state, update algorithm, as well as the concept of time, is changed.  As in the original I2A paper, they experiment with learning how best to take advantage of a learned dynamics model.

To me, this strain of work is very important.  Because no model is perfect, I think the idea of learning how best to use an approximate model will become increasingly important.  The original I2A of learning-to-interpret models is here extended with the idea of learning-to-query, and a variety of solid variants on a few basic themes are well worked out.

Overall, I think the strongest part of this paper is in its conceptual contributions - I find the work thought-provoking and inspiring.  On the negative side, I felt that the experiments were thin, and that the work was not well framed in terms of the literature of state-space identification and planning (there are a zillion ways to plan using a model; couldn't we have compared to at least one of them?  Or discussed a few popular ones, and why they aren't likely to work?  Since your model is fully differentiable, vanilla MPC would be a natural choice [In other words, instead of learning a rollout policy, do something simple like run MPC on the approximate model, and pass the resulting optimized action trajectory back to the agent as an input feature]).

Of course, while we can always demand more and more experiments, I felt that this paper did a good enough job to merit publication.

Minor quibble: I wasn't sure what to make of the sSSM ideas.  My understanding is that in any dynamical system model, the belief state update is always a deterministic function of previous belief state and observation; this suggests to me that the idea of "state" here differs from my definition of "state".  I don't think you should have to sample anything if you've represented your state cleanly.

---

> ### Author Response · Authors · 2017-12-14
> **Re: Could be impactful**
>
> We thank the reviewer for the comments.
>
> "... I felt that the experiments were thin, and that the work was not
> well framed in terms of the literature of state-space identification and
> planning"
>
> We agree that a baseline which uses the learned models together with a
> classical planning algorithm, would be very informative and we are currently
> working on implementing MPC on a continuous relaxation of the discrete action
> model. However, we expect this baseline to be very slow, as the cost of
> backpropagating through the model is high; we expect having an optimization as
> an inner loop of the agent, will make this at least 10x slower at test time
> compared to the imagination-augmented agent.
>
> "Minor quibble: I wasn't sure what to make of the sSSM ideas.  My understanding
> is that in any dynamical system model, the belief state update is always a
> deterministic function of previous belief state and observation; this suggests
> to me that the idea of "state" here differs from my definition of "state".  I
> don't think you should have to sample anything if you've represented your state
> cleanly."
>
> This is a very interesting and subtle point. It is true, that the *ideal* belief-state
> updated is a deterministic function of the previous belief-state and the current
> observation. Given a simple  model, eg linear-gaussian dynamical systems or HMM,
> optimal inference can and actually is done in a deterministic way (ie Kalman filter, Viterbi
> algorithm). However, the expressive power of these models are too limited for the
> ALE domains. In more powerful, non-linear models as we consider here, we have to
> approximate inference, and this can be done in multiple ways: trying to
> approximate the entire belief-state, or use a single sample, as we have done
> (other approaches also look at multi-sample particle filtering). It's very much
> an open question what the best approximation strategy is given the same *finite
> resources* (e.g. neural network with n layers). We build on the sample-based
> literature, which has achieved good results in sequence modelling such as speech
> etc (see the generative modelling references in the main article), but for us this is a decision of algorithmic simplicity / performance, not of dogma.

---

### Public Comment · (anonymous) · 2018-01-22
**Can authors justify use of "dynamic" in the title?**

Having gone through both this paper and I2A paper which this builds on, I find that the claim that any sort of "dynamic" state abstractions is learned to be unjustified.

I find this behaviour of using overly generic titles that don't hold any water to flag plant alarming.

---

> ### Author Response · Authors · 2018-01-24
> **re: "Can authors justify use of "dynamic" in the title? "**
>
> Would the author of the comment elaborate on their objection?
>
> The title is justified in our opinion; we use the term "dynamic state abstraction" to emphasize the following:
> - we learn state presentations that are more compact than the the raw  observations at a single time step, hence they constitute "abstractions".
> - these representations are learned by predicting future observations , hence they capture the dynamics of the environment
> - we experimentally show that these learned state representations (together with the learned transition function) contain sufficient information to accurately predict the future over tens to hundreds of raw frames in non-trivial environments.

---

> > ### Public Comment · (anonymous) · 2018-01-29
> > **re: "Can authors justify use of "dynamic" in the title? "**
> >
> > Learning abstractions of environment dynamics and learning dynamic abstractions are entirely different things. One implies stationarity of the abstraction hierarchy and the other does not.

---

### Decision · Program_Chairs · 2018-01-29
**ICLR 2018 Conference Acceptance Decision**

**Decision:**

Reject

**Comment:**

There was quite a bit of discussion about this paper but in the end the majority felt that, though the paper is interesting, the results are too limited and more needs to be done for publication.

PROS:
1. Good comparison of state space model variations
2. Good writing (perhaps a bit dense in places)
3. Promising results, especially concerning speedup

CONS:
1. The evaluation is quite limited